# Willingness intensity and co-evolution of decision rationality depending on aspiration enhance cooperation in the spatial public goods game

**Shounan Lu[1], Ge Zhu[2], Jianhua Dai [3]***

**1** College of Science, Beijing University of Posts and Telecommunications, Beijing, China, **2** School of Information Management, Beijing Information Science & Technology University, Beijing, China, **3** Business School, China University of Political Science and Law, Beijing, China

* daijianhua@cupl.edu.cn

**Data Availability Statement:** All relevant data are within the paper and its Supporting Information files.

## Abstract

The Fermi rule states that rational or irrational sentiment affects individual decision-making. Existing studies have assumed that individuals' irrational sentiments and behavior willingness have fixed values and do not change with time. In reality, people's rationality sentiment and behavior willingness may be influenced by some factors. Therefore, we propose a spatial public goods game mechanism, in which individuals' rational sentiment is co-evolution synchronously depending on the difference between aspiration and payoff. Moreover, the intensity of their subjective willingness to change the status quo depends on the gap between aspiration and payoff. We likewise compare the combined promotion effect of the stochastic "Win-Stay-Lose-Shift" (*WSLS*) and random imitation (*IM*) rules. Simulation experiments indicate that high enhancement factors are not conducive to cooperation under the *IM* rules. When aspiration is small, *WSLS* is more conducive to promoting cooperation than *IM*, while increasing aspiration, and the opposite phenomenon will appear. The heterogeneous strategic update rule is beneficial to the evolution of cooperation. Lastly, we find that this mechanism performs better than the traditional case in enhancing cooperation.

## Introduction

Cooperation plays an important role in promoting progress in real social and biological systems. However, Darwin's theory of evolution [1, 2], which contradicts the emergence and maintenance of cooperative behavior among selfish individuals, indicates that exploring the emergence and maintenance of cooperative behavior among selfish individuals is among the key and interesting issues [3, 4]. The public goods game (PGG), as a typical paradigm of the multiplayer game model, is widely concerned in this field [5, 6]. In PGG, $N$ individuals independently and simultaneously decide to cooperate or not. Cooperators will contribute 1 to public pools, while defectors will contribute nothing. The sum of all contributions is multiplied by the enhancement factor $r > 1$. The amount is redistributed equally among all individuals in

**Funding:** This work was supported by national key research and development plan of China under grants nos. 2019YFB1406500 and the talent introduction funds of China University of Political Science and Law. The funders had no role in study design, data collection and analysis, decision to publish, or preparation of the manuscript.

the whole group, regardless of collaborators or defectors. In PGG, defection is the optimal strategy, no matter other people's strategies, which is a common phenomenon in social and biological systems, such as the global climate problem [7, 8] and public traffic [9]. The emergence and maintenance of cooperation remains a puzzle [10, 11]. Combating with evolutionary game theory [1], which provides a theoretical framework for analyzing the emergence and maintenance of cooperative behavior among selfish individuals [12, 13].

Many scholars have proposed different mechanisms to explain the reasons for the emergence and maintenance of cooperation. Nowak [14] summarized five main mechanisms conducive to cooperation, namely, kin selection, direct reciprocity, indirect reciprocity, network reciprocity (or spatial reciprocity), and group selection, which can effectively promote cooperation. Given that social relations have a certain spatial structure in real social systems, network reciprocity particularly elaborates that a structured population is beneficial to the prevalence of cooperative behavior. Relevant studies have also confirmed that cooperation can be enhanced in a specific network structure [15–18]. Depending on network reciprocity, various cooperation promotion mechanisms have been proposed, such as reputation [19, 20], optional participation [21, 22], teaching activity [23, 24], punishment and incentive [25–29], and other novel mechanisms and methods [30, 31].

In recent years, individuals' aspiration as an important endogenous feature has become a significant concern. Simon [32, 33] used individuals' expectations of future revenues as bases in defining aspiration level as a value of a goal variable that must be reached or surpassed by a satisfactory decision alternative, rather than pursuing the most profitable strategy, based on the hypothesis that humans are bounded rationally. In reality, everyone has aspirations or expectations, which are related to the environment, economy, and individual production efficiency. When their income is less than their aspirations or expectations, they will tend to pursue some measures to change the status quo [34–36]. Some studies have reported the impact of aspiration on cooperation. Shen et al. [37] analyzed the relationships between aspiration and cooperation based on the closeness of relationship among individuals. They found that aspiration can promote cooperation. Liu et al. [38] studied cooperative behavior depending on the "win–stay–lose–shift" (WSLS) strategy update rule based on PGG. They also indicated that a moderate aspiration level will enhance cooperation. However, the preceding studies assumed that all individuals have a fixed and undifferentiated rationality value in the decision-making process [23, 37–40], and they do not consider the effect of heterogeneity or other factors (i.e., cognitive differences) on rationality.

In real social systems, people's rationality levels are heterogeneous. People's rational sentiments are affected by various factors, such as incomplete information or personal sentiment [41], and interaction and learning environments [42]. Individuals' rational sentiments play important roles in behavioral decisions [43, 44]. Wang et al. [45] explained that interfering factors (e.g., rational sentiment) in the decision-making process would affect the evolution of cooperation. Decision deviation will also occur at different rational levels. For example, the Fermi rule [46] often considers that completely rational individuals constantly adopt the best strategies to pursue maximum payoffs. However, some irrational individuals have a small possibility of selecting the worse ones. When individuals are completely irrational, their behavior is easily disturbed by random decision-making. Although some studies have reported that moderate irrationality can promote cooperation [47, 48], existing research has assumed that people's decision behavior is at the same rational level even in a heterogeneous population, which is inconsistent with reality [49–52]. In real social production activities, changes in the environment and individuals' cognitive abilities with the passage of time will lead to synchronous changes in information processing abilities and rationality levels. Kahn et al. [53] indicated that individuals' last experiences will have an impact on their future behavior. A

common and easily understandable example is that continuous failure to achieve aspirations may lead individuals to consider various factors to update their strategies. They will specifically prioritize their own earnings and become markedly rational. By contrast, continuously achieving aspirations may lead individuals to disregard some factors in updating their irrational strategies [54, 55]. Inspired by this situation, we propose a spatial PGG mechanism, in which individual rational sentiments synchronously change based on aspirations. That is, satisfaction with current payoffs will affect individuals' rational decision-making levels in the next round. In addition, we are convinced that the intensity of individuals' subjective willingness to change the status quo is influenced by the gap between aspiration and payoff. That is, the greater the gap, the stronger the desire to change (or not change) the status quo. We emphasize that we only consider the benefits of individuals in the local environment rather than the cumulative benefits from neighbor groups. Furthermore, we consider the critical role of the strategy updating rules in the evolution of cooperation. We compare the impact of two types of strategy updating rules on cooperation: aspiration-driven imitation (*IM*) rule and aspiration-driven stochastic *WSLS* rule. Simulation and comparison yield some valuable results. Higher enhancement factor is not conducive to the formation of cooperation under *IM*. Small aspiration level is more conducive to promoting cooperation under *WSLS* than *IM*, while increasing aspiration results in the opposite. Lastly, the heterogeneous strategic update rule will benefit from the evolution of cooperation. Our research compensates for the shortcomings of existing studies and will further help us understand the emergence and maintenance of cooperation.

## Model

We consider PGG located on a two-dimensional square lattice network with periodic boundary conditions. Each node on the square lattice network represents an individual. The focus individual interacts with the four nearest neighbors from a group with size $N_g = 5$ and adjusts the corresponding strategies after each generation of the game. In the initial stage, all players are divided into cooperators (*C*) or defectors (*D*) with equal probability. In each generation, cooperators contribute $S_i = 1$ to the public pool and defectors contribute nothing. Returns of focus individual *i* depend on its strategy $S_i$ and the number of cooperators $N_C$ in the group game $N_g$. Thus, individual *i*'s payoff can be described as Eq (1), and each one obtains payoffs the same way. is the number of cooperators in the group game g:

$$P_i = \frac{r \times (N_c + S_i)}{N_g} - S_i \qquad \text{Eq(1)}$$

where *r* represents the enhancement factor. If player *i* is a cooperator, then $S_i$ equals 1; otherwise, $S_i$ is 0.

We follow the existing research [53] and introduce parameter *A* to quantify individuals' aspiration levels. For PGG, which is a group game with multi-player participants, players' payoffs depend on the number of cooperators in the group with size $N_g = 5$. Therefore, the more individuals in the group contributing to the public pool, the more payoffs the focus individuals will expect. Given that individuals' payoffs are linearly related to the number of cooperators in the group, we conclude that the relationship between individuals' expected payoffs and *A* is linear. Individual *i*'s expected payoff can be described as Eq (2).

$$P_{iA} = k_i A, \qquad \text{Eq(2)}$$

where $A \in [0,1]$ refers to quantifying individuals' aspiration levels, note that *A* is fixed and will not change during the evolutionary process. and $k_i = 5$ represents the number of people involved in investing in public pools.

After completing a game round and before introducing the strategy update rules, we should introduce the iteration mechanism of individuals' rational level $R_i$ and the determination method of individuals' desire $w_i$ to update the strategy. The preceding discussion indicates that when individuals achieve expectation payoffs in the current round, they will disregard some factors to update their strategies in the next round and their rationality levels will decrease, some of which will become irrational. Otherwise, they will consider various factors to earn additional payoffs in the next round, thereby increasing their rationality levels. After each game round, all individuals will determine their willingness to adjust strategies depending on the gap between aspiration and expected return. The existing literature has assumed that players' increase or decrease in willingness is a fixed value in each *MCS* regardless if the actual income is greater (lesser) or considerably greater (lesser) than the aspiration [56]. Moreover, they disregard the subjective desire intensity to which individuals' willingness is affected caused by the gap between actual income and aspiration. In reality, the larger the gap between income and aspiration, the stronger the willingness of individuals to adjust strategies (or not), instead of increasing or decreasing a fixed value. For example, when individuals are extremely dissatisfied with their returns, they will have a strong desire to change the status quo, and vice versa. The iterative rules of $R$ and $w$ are as follows.

a) Individuals' rationality level $R$. Given the heterogeneity among different individuals, when $R_i = 0$, individual $i$ is completely rational. When $R_i > 0$, individual $i$ is relatively irrational. When $R_i \rightarrow \infty$, individual $i$ is completely irrational. We let the range of individuals' rationality level to be $R \in [0, 2]$, and $\beta$ is a fixed value equal to 0.05.

$$R(t+1) = \begin{cases} R(t) + \beta \; if \; P_i > P_{iA} \\ R(t) - \beta \; if \; P_i < P_{iA} \end{cases} \qquad \text{Eq(3)}$$

b) Individuals' willingness to update strategy (or satisfaction with payoff) $w$. All individuals will adjust their willingness to update depending on the gap between income and aspiration. To avoid individuals falling into the dilemma of not adjusting strategies, we set the range of willingness to update at $w \in [0, 1]$.

$$w_i = tanh(P_{iA} - P_i) \qquad \text{Eq(4)}$$

We introduce the strategy update mechanism. All players will adjust their strategies independently and simultaneously after each round of the game with their nearest neighbors. To explore more comprehensively the influence of evolutionary PGG, depending on aspiration-driven, we consider two typical updating rules.

a) The stochastic WSLS strategy updating rule depends on the Fermi function. Individual $i$ will change his/her current strategy into the opposite base on probability $H_i$ when dissatisfied with current payoffs. When his/her payoff is greater than aspiration, he/she will be satisfied with the status quo and unwilling to make changes, resulting in lower probability $H_i$ for individuals to adjust their strategies.

$$H_i = \frac{w_i}{1 + \exp[(P_i - P_{iA})/R_i]}, \qquad \text{Eq(5)}$$

where $P_i$ and $P_{iA}$ are individual $i$'s payoff and expected payoff, respectively; and parameters $w_i$ and $R_i$ depict individuals' willingness to update strategies and rationality levels, respectively.

b) IM based on payoff difference depend on the Fermi function. When individual $i$ is dissatisfied with the payoff, center individual $i$ will have a higher willingness to change his/her strategy determined by $w_i$, randomly select an individual $j$ from four nearest neighbors, and whether or not to imitate individual $j$'s strategy based on probability $W$. The more the $w_i$, the more the $W_{(Si \leftarrow Sj)}$. The basic consensus is that individual $i$ will keep his/her current strategy when satisfied with the current payoff.

$$W(S_i \leftarrow S_j) = \frac{w_i}{1 + \exp[(P_i - P_j)/R_i]}, \qquad \text{Eq(6)}$$

where $P_i$ and $P_j$ are individual $i$ and $j$'s payoffs, respectively; $S_i$ and $S_j$ are individual $i$ and $j$'s strategies, respectively; and parameters $w_i$ and $R_i$ depict individual $i$'s willingness to update strategy and rationality level, respectively.

To evaluate the promotion effect of the preceding mechanisms on cooperation, we simulate the evolutionary game on $N = 100 \times 100$ square lattices with periodic boundaries in accordance with the Monte Carlo simulation process. Initially, strategies of $C$ and $D$ are randomly distributed among all individuals with equal probability ($p = 0.5$). Individuals have equal chances of adjusting their strategies in each full Monte Carlo step (*MCS*). Key parameters' frequency of cooperators $fc$ in steady state lasts $1.1 \times 10^4$ *MCS*. To remove the unstable part anterior and make the result as accurate as possible, we take the average cooperation rate of the stable phase (i.e., last $10^3$ steps).

## Simulation and analysis

We initially study the impact of aspiration based on the co-evolution of willingness to update strategies and rationality levels on the evolution of cooperation for different enhancement factor $r$. Fig 1 shows the frequency of cooperators $fc$ dependent on aspiration $A$ under *WSLS*, which has an appropriate aspiration $A$ leading to the highest cooperation frequency $fc$ for a fixed enhancement factor $r$. From a global perspective, this phenomenon is consistent with the literature [38], in which significantly high or low aspirations are not conducive to cooperation. Moreover, increasing reinforcement factors will mitigate the negative impact of high aspiration on cooperation. For example, when $r = 5$, $fc = 54.26\%$, and increasing $r = 6$, cooperation

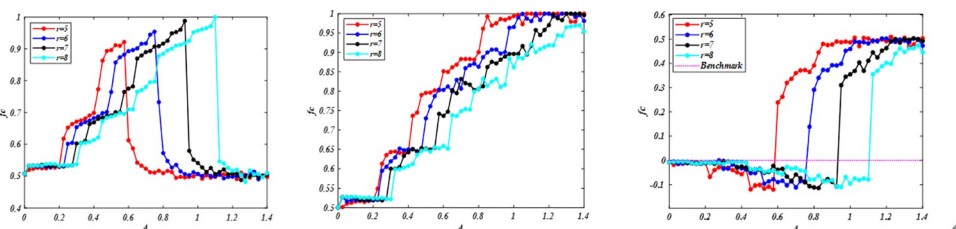

**Fig 1. Frequency of cooperation $fc$ is dependent on $A$ for different values of enhancement factor $r$.** Note that different enhancement factor $r$ corresponds to different optimal aspiration level $A$ in the evolution of cooperation. (a) evolution of cooperation behavior under the stochastic *WSLS* strategy updating rule; (b) evolution of cooperative behavior under the *IM* rule; (c) gap of cooperative frequency between *IM* and *WSLS* under the same condition; when $fc_{(IM-WSLS)}$ is above 0, *IM* has a better promotion effect on cooperation than *WSLS* under the same condition; otherwise, *WSLS* can better promote cooperation than *IM*; benchmark line is used to show whether it is above or below 0. All results are obtained under $L = 100$. (a) *WSLS*, (b) *IM*, and (c) The gap of $fc$.

level will increase $fc$ = 80.87%. From a local perspective, we can observe a discontinuous phase transition phenomenon, in which the effect of aspiration $A$ on cooperation is divided into different regions. In the same region, cooperation level has a slight difference compared with the cooperation level corresponding to critical value $A$. For example, when $0.3 < A < 0.475$ for a fixed $r$ = 7, the range of cooperation level is from 65.37% to 69.78%, with small fluctuations. By decreasing aspiration to $A$ = 0.35, cooperation level will evidently decrease to a minimally cooperative stable state at approximately $fc$ = 60.44%. By increasing aspiration to $A$ = 0.475, cooperation level will evidently increase to a markedly cooperative stable state at approximately $fc$ = 76.43%. Similar phenomena can be found under the *IM* rule, including but not limited to discontinuous phase transition. Moreover, the difference is that high aspirations will enhance cooperation. The promotion effect of *IM* on cooperation is weaker than that of *WSLS* with low-level aspiration. By increasing aspiration, *IM* has more advantages in promoting cooperation than WSLS. However, the advantage will be reduced with enhancement factor $r$. Thus, appropriate ambition can promote cooperation under the *WSLS* rule. In the *IM* and *WSLS* rules, improving the enhancement factor is not conducive to cooperation, which is different from the conclusions of previous studies [57, 58]. We also found that the promotion effect of *IM* on cooperation is more effective than that of *WSLS* under high aspiration.

To explain the counter-intuitive behaviors fc depending on r that high enhancement factor $r$ results in lower cooperation level under certain conditions, we investigate the transition probabilities of $C$ players changing into $D$ players $W_{C \to D}$, and $D$ players changing into $C$ players $W_{D \to C}$ as functions of $A$ for different $r$. When the system has reached steady state, we can have the number of cooperators that change to defectors equal to the number of the defectors that change to cooperators. Hence, we can have $N(1-fc)\ W_{D \to C} = NfcW_{C \to D}$ approximatively by the mean-field techniques [34]. Therefore, cooperation level can be written as $fc = 1 + W_{C \to D}/W_{D \to C}$. By using this equation, we can analyze $fc$ with respect to $A$ or r by using the probabilities $W_{C \to D}$ and $W_{D \to C}$.

As shown in Fig 2 under *WSLS* rule, as aspiration increases, defectors and cooperators are dissatisfied with their strategies and update their strategies more frequently, which is depicted by the high transition probabilities of $W_{C \to D}$ and $W_{D \to C}$. By increasing enhancement factor r, cooperators and defectors' payoffs are improved, although such an improvement is insufficient to realize their aspiration. They still frequently update their strategies, resulting in a decrease in cooperation level.

As shown in Fig 2 under IM rule, increasing enhancement factor r results in an increase in payoffs of cooperators and defectors and decrease in the probability of changing their strategies, thereby leading to a reduction in the system cooperation level.

We analyze the time evolution of cooperation for different parameters. Fig 3 present the evolution of cooperation for different aspiration $A$ under a fixed enhancement factor $r$. Note that the frequency of cooperators $fc$ gradually increases in the initial stage and eventually reaches a steady state. When aspiration increases from $A$ = 0.3 to A = 0.5, the system gradually increases to a markedly cooperative steady state, cooperation level will increase from $fc$ = 65.37% to $fc$ = 76.43% for a fixed enhancement factor $r$ = 6, and the system will increase to a substantially cooperative steady state. Moreover, as aspiration increases from $A$ = 0.5 to $A$ = 0.7, the system continues in a markedly cooperative steady state. However, when aspiration increases to $A$ = 0.9, the system will continue to a minimally cooperative stable state. Cooperation level decreases from $fc$ = 91.71% to $fc$ = 51.49%. That is, there is a moderate aspiration $A$ leading to the highest frequency of cooperator $fc$ for a given enhancement factor $r$. Accordingly, phase transition will occur from A = 0.7 to A = 0.9. Subgraph (b) shows the time evolution process of cooperation on a high enhancement factor. Increasing $r$ = 6 to $r$ = 7 leads to increases in the critical value of aspiration $A$ for phase transition, leading the system to a

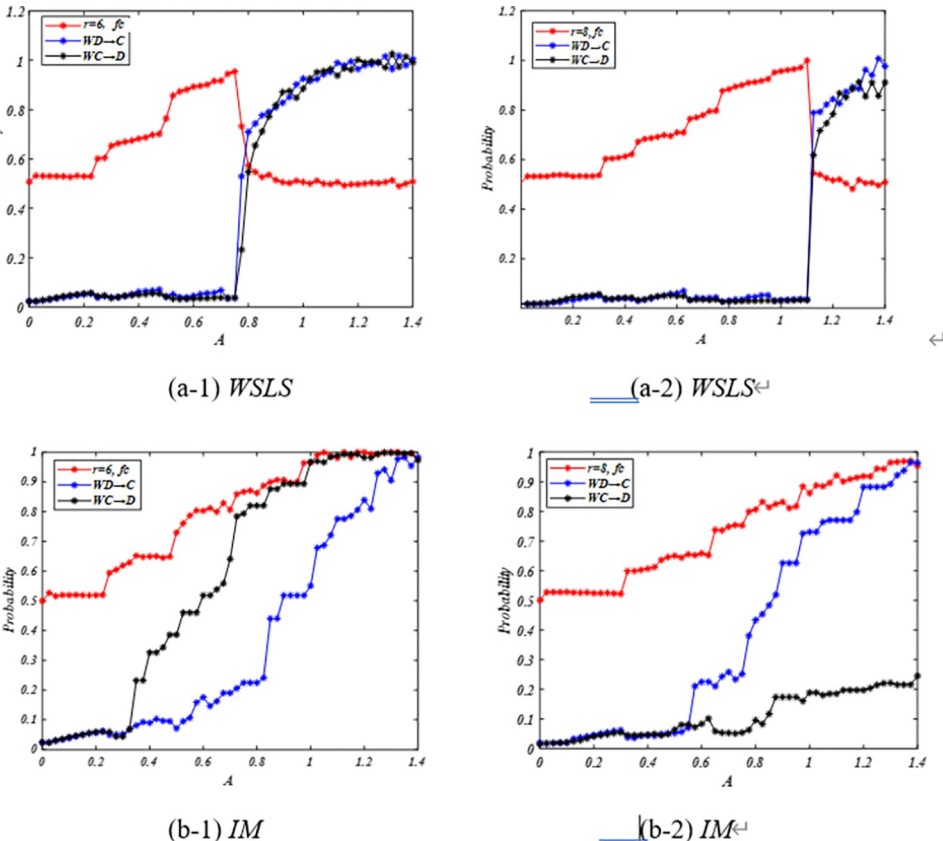

**Fig 2. Transition probabilities and cooperation level as functions of aspiration A depend on *WSLS* on the regular lattice for *r* = 6 (panel (a-1)) and *r* = 8 (panel (a-2)); *r* = 6 (panel (b-1)) and *r* = 8 (panel (b-2)) depend on *IM*.** Red denotes the cooperation level *fc*, blue denotes the transition probability of *D* players changing into *C* players $W_{D \to C}$, and black denotes the transition probability of *C* players changing into *D* players $W_{C \to D}$.

markedly cooperative stable state when *A* = 0.9. Subgraph (c) shows the time evolution process of cooperation under the *WSLS* and *IM* rules. Note that the frequency of cooperators *fc* gradually rapidly increases in the initial stage and eventually reaches a steady state rapidly under *IM*. When A = 0.9 (r = 6), cooperators more easily survive under the *IM* rule than the *WSLS* rule. When increasing *r* = 7, cooperation density based on the *IM* rule is larger than that based on the *WSLS* rule. Therefore, under the *WSLS* rule, appropriate aspiration can promote cooperation, and increasing the enhancement factor can increase the phase transition point of

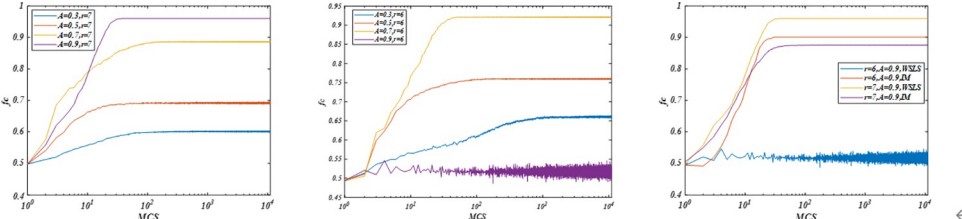

**Fig 3. Time evolution of cooperation frequency of cooperation *fc* as a function of *MCS* for different parameters.** (a) Evolution process for *A* = 0.3, 0.5, 0.7, and 0.9, and *r* = 6; (b) Evolution process for *A* = 0.3, 0.5, 0.7, and 0.9, and *r* = 7; (c) Evolution process for *A* = 0.9 and *r* = 6, 7, under IM and WSLS, respectively. All results are obtained under *L* = 100. (a) *WSLS*, (b) *WSLS*, and (c) *WSLS-IM*.

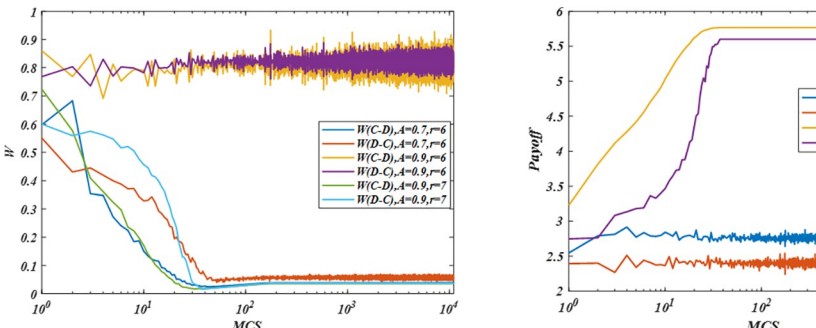

**Fig 4. Transition probability and payoff as a function of *MCS* for different parameters.** (a) transition probability under aspiration *A* = 0.7, 0.9 and *r* = 6, 7; (b) Payoff of cooperators under aspiration *A* = 0.9 and *r* = 6, 7. All results are obtained under *r* = 7 and *L* = 100 based on the *WSLS* rule.

aspiration's influence on cooperation, thereby further promoting cooperation. Under the *IM* rule, increasing the enhancement factor can also increase the phase transition point of aspiration but inhibit cooperation.

We further analyze the transition probability based on the mean-field techniques [56, 59] and payoffs in Fig 4, and we found that cooperators can obtain more payoff than defectors, as shown in subgraph (a). The probability of cooperators changing into defectors is smaller than defectors changing into cooperators. Under this condition, defectors gradually turn into cooperators to obtain higher returns, thereby leading the system to a more stable state, as shown in subgraph (b). However, when aspiration is considerably high for a given enhancement factor, cooperators and defectors are dissatisfied with the returns, and they have a high willingness to change their strategies. As shown in subgraph (b) and (c), cooperators and defectors begin to change strategy frequently and blindly. Cooperators have difficulty forming stable and large clusters. Meanwhile, cooperators and defectors' payoffs fall to a considerably low level. Lastly, strategies are approximately distributed uniformly among the whole population, as shown in Fig 5. A detailed explanation for the high aspirations is not conducive to the formation of cooperation.

To further understand the emergence and maintenance of cooperation among the whole population, the strategic distribution of populations should be analyzed from a micro perspective.

Fig 5 shows the distribution of the population's strategies in the time evolution process for several time steps from a micro perspective. When the aspiration is small, *A* = 0.3, both cooperators and defectors are satisfied with returns, as shown in Fig 5, they have a lower willingness to change the status quo. However, there are still a few individuals close to the cooperators cluster for a free ride on the whole population. Moreover, cooperators form clusters to resist the invasion of defectors by obtaining stable benefits. Clusters gradually expand with an increase in aspiration. Moreover, by analyzing the distribution of cooperation strategy and the corresponding rational emotional state of individuals in the stable state for different aspirations *A* in snapshots (c), completely rational individuals always maintain a high willingness to update their strategies (in snapshots (b)), that is, they always hope to maximize their own benefits by constantly adjusting their strategies in which cooperation is difficult to maintain. The reasonable explanation that some rationality is conducive to cooperation is that the decision-making process of individuals not only considers the maximization of benefits, but also considers the external environment and decision-making conditions, which leads to the maintenance of cooperation.

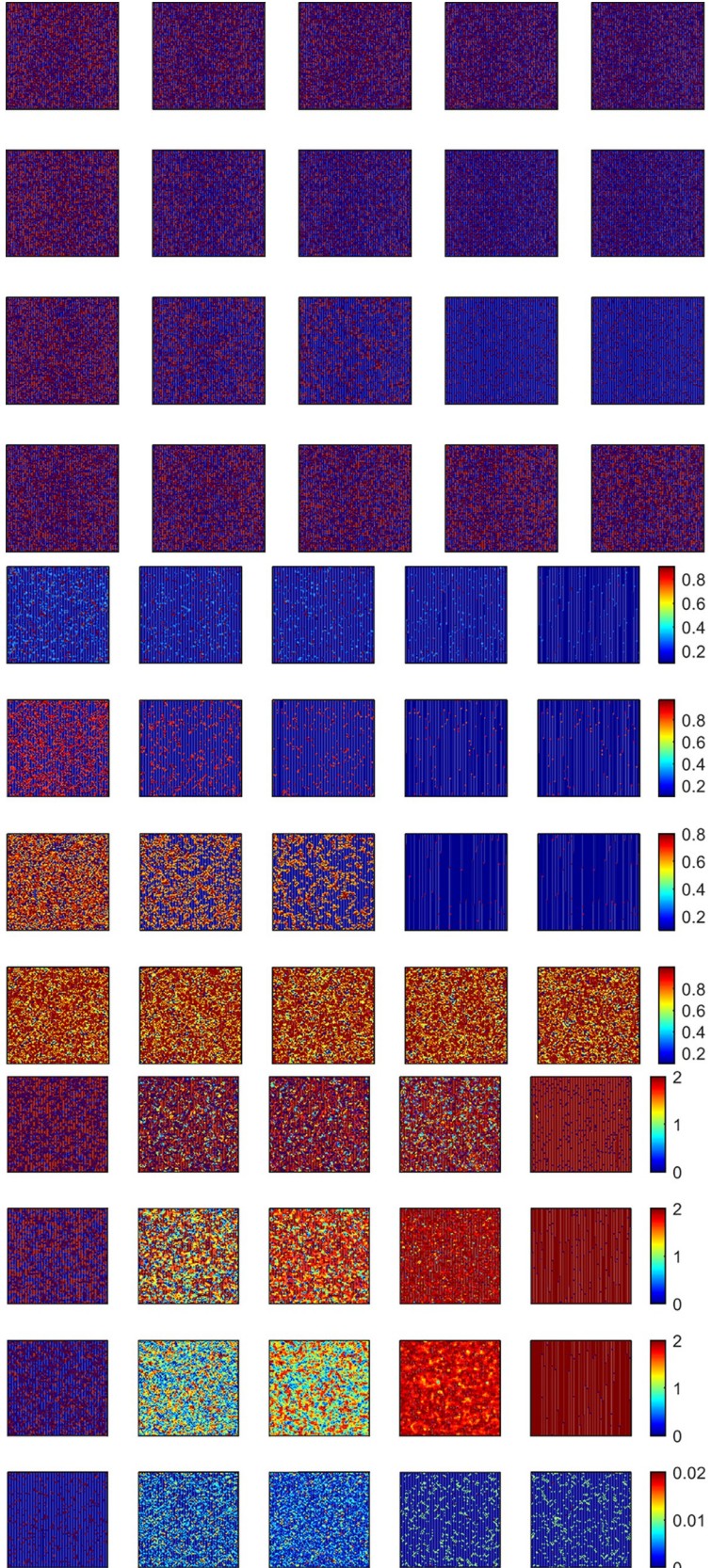

**Fig 5.** Snapshots (a) is snapshots of cooperation (blue) and defection (purple) strategy distribution, and snapshots (b-c) is willingness to update and rational distribution on the two-dimensional lattice network for some *MCS*. From top to bottom, the values of *A* = 0.3, 0.5, 0.7, and 0.9. Snapshots were taken at *MCS* = 1, 5, 10, 100, and 11000 from left to right. All results are obtained under *L* = 100 and *r* = 6 based on the *WSLS* rule.

When aspiration is small, the population's behavior is similar between *r* = 6 and *r* = 7. However, compared with aspiration *A* = 0.9 under enhancement factors *r* = 6, increasing the enhancement factors enhances *r* = 7, cooperation is shown in Fig 6, thereby maintaining the population at a high cooperation level again. Similar phenomena also indicate that certain rationality can promote cooperation in Fig 7. We further analyze transition probability and payoffs and determine that increasing aspiration makes individuals dissatisfied with the status quo. Moreover, improving the enhancement factor will make individuals obtain satisfactory returns again. For cooperation, they obtain more payoffs than defections. As the game advances, the system leads to a markedly cooperative stable state, and most of them have low willingness to change the status quo. The microscopic description of cooperation is consistent with that shown in Fig 1.

Compared with the *WSLS* rules, when population evolution depends on the *IM* rule, under the condition of *A* = 0.9 (*r* = 6), cooperators gradually form clusters to resist the invasion of defectors with the passage of time. Cooperators can obtain stable returns by forming clusters, eventually maintaining a stable state shown in Fig 8. What's more, by comparing the level of rational emotional state and renewal willingness in Fig 9, some completely rational individuals have a lower strategy renewal willingness in a stable state. This means that individuals are

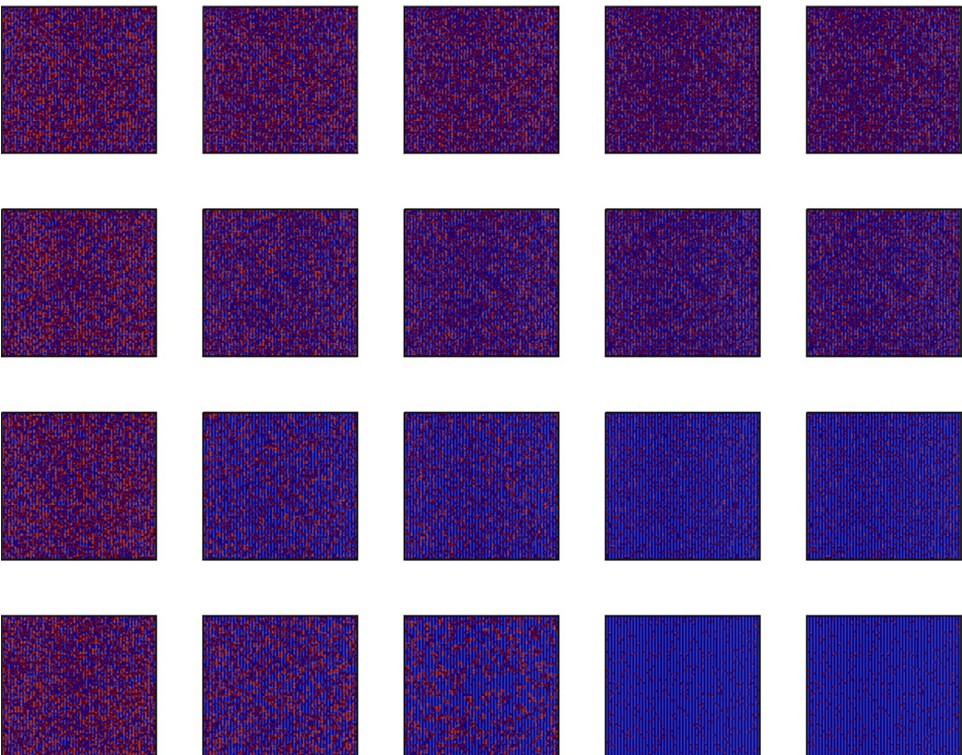

**Fig 6. Snapshots of cooperation (blue) and defection (yellow) strategy distribution on the two-dimensional lattice network for some *MCS*.** From top to bottom, *A* = 0.3, 0.5, 0.7, and 0.9, and *r* = 7. Snapshots were taken at *MCS* = 1, 5, 10, 100, and 11000 from left to right. All results are obtained under *L* = 100 based on the *WSLS* rule.

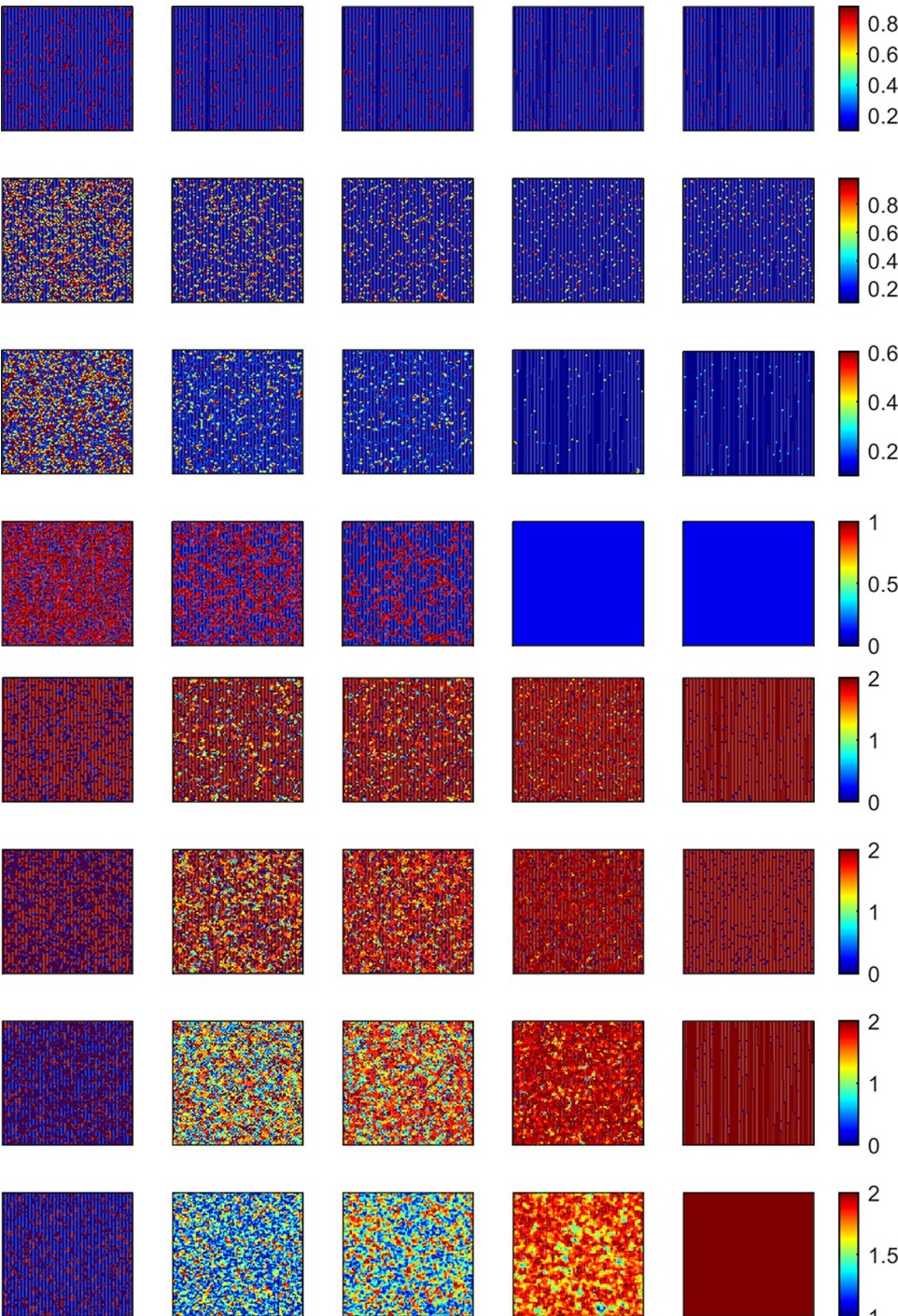

**Fig 7. Snapshots of willingness to update and rational distribution on the two-dimensional lattice network for some *MCS*, *w* = [0.1,1].** From top to bottom, *A* = 0.3, 0.5, 0.7, and 0.9, and *r* = 7. Snapshots were taken at *MCS* = 1, 5, 10, 100, and 11000 from left to right. All results are obtained under *L* = 100 based on the *WSLS* rule. (a) Willingness distribution and (b) Rational distribution.

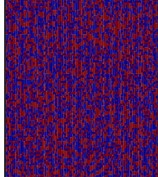 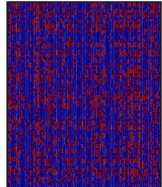 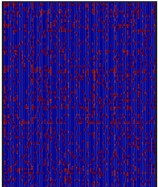 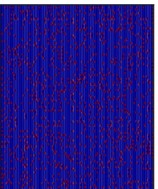 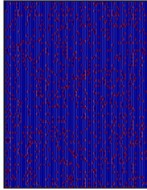

**Fig 8. Snapshots of cooperator (blue) and defector (purple) strategies on the regular lattice depending on IM, *A* = 0.9 and *r* = 6. Snapshots were taken at *MCS* = 1, 5, 10, 100, and 11000 from left to right.** All results are obtained under *L* = 100 based on the *IM* rule.

more likely to meet the given ambition value by adjusting strategies based on imitation rules, and the lower intention of updating strategies reduces the probability of individuals being invaded by traitors, which is conducive to the maintenance of cooperation. Therefore, imitation rules are more conducive to the emergence and maintenance of cooperation under specific conditions.

To further explain the reason why large-scale cooperation still exists under the condition of high aspiration, we consider the strengthening effect of heterogeneous populations on cooperation. According to the strategy adjustment rules, we divide individuals into two categories: Types A and B adjust strategies depending on the *IM* and *WSLS* rules, respectively, in which players adjust their respective types based on Eq 6. As shown in Fig 10, the mixed update rules promote cooperation compared with the pure *IM* or *WSLS* rule. When *A* = 0.5 (*r* = 6), the population is almost occupied by imitation rules. With an increase of *A* = 0.6 (*r* = 6), the rules coexist in whole population. With an increase of *A* = 0.7 (*r* = 6), the population is almost occupied by the WSLS rules. Therefore, the hybrid strategy update rules can promote cooperation.

Lastly, to reflect that the co-evolutionary mechanism is more effective in strengthening cooperation than the traditional situation (or traditional case) without any mechanism, we compared this mechanism with the traditional situation, as shown in Fig 11. Whether IM or WSLS, the co0evolutionary mechanism performs better than the traditional situation.

## Conclusion

In an actual social system, people may not constantly make decisions under rational or irrational sentiments. They are easily influenced by such as experiences of failures or successes and cognitive abilities. Moreover, people's subjective willingness to adjust strategies is often determined by their satisfaction with payoffs. For example, frequent success will increase people's

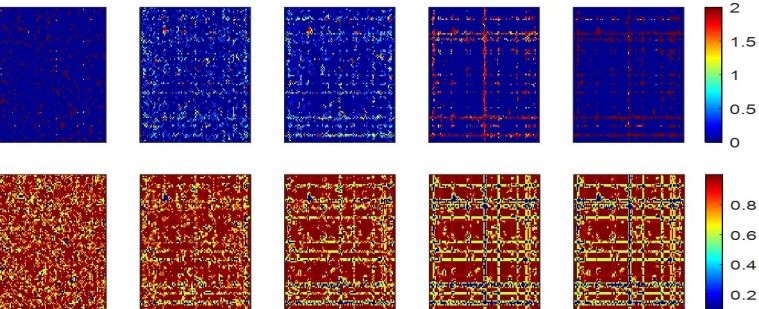

**Fig 9. Snapshots of willingness and rationality distribution on the regular lattice depending on *IM*, *w* = [0.1,1]. *A* = 0.9 and *r* = 6. Snapshots were taken at *MCS* = 1, 5, 10, 100, and 11000 from left to right.** All results are obtained under *L* = 100 based on the *IM* rule.

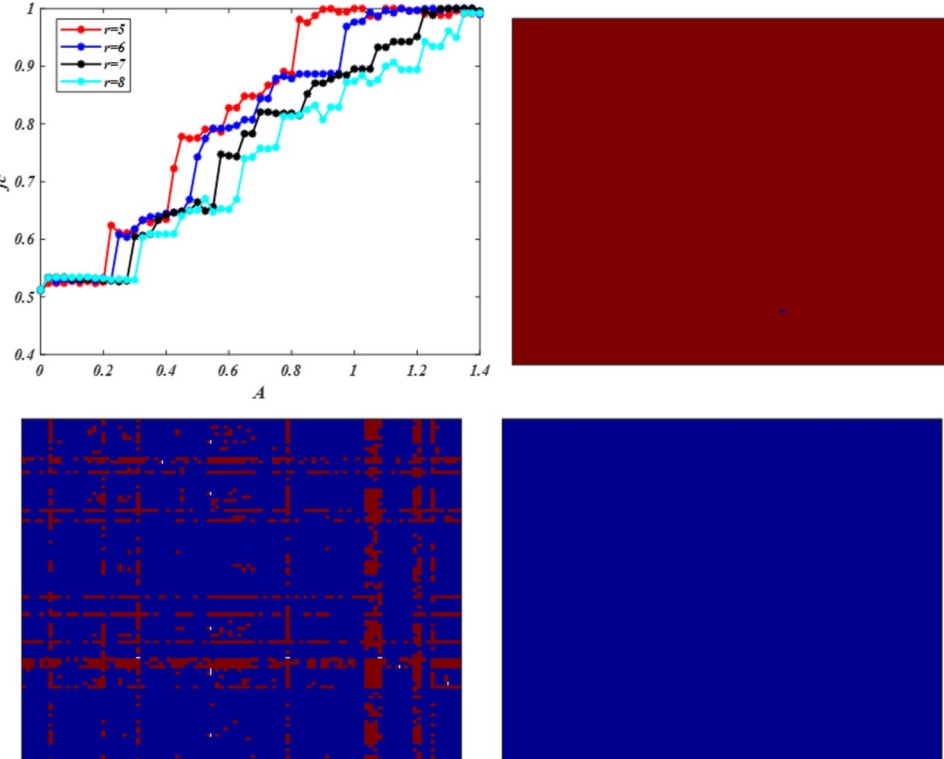

**Fig 10.** (a) Frequency of cooperation *fc* is dependent on *A* for different values of enhancement factor *r* under hybrid strategy updates rules; (*b-d*) Snapshots of types *A* (purple) and *B* (blue) on the regular network.

irrational sentiments. By contrast, failure to achieve aspirations consistently may lead individuals to consider various factors to earn more payoffs and become more rational. Meanwhile, if their income is substantially less than their expectations, then they will have a stronger willingness to change the status quo, and vice versa. Inspired by this phenomenon, we propose a co-evolution mechanism of aspiration-driven rational sentiments. Through numerous simulations, we can observe that different aspiration and enhancement factors have different optimal

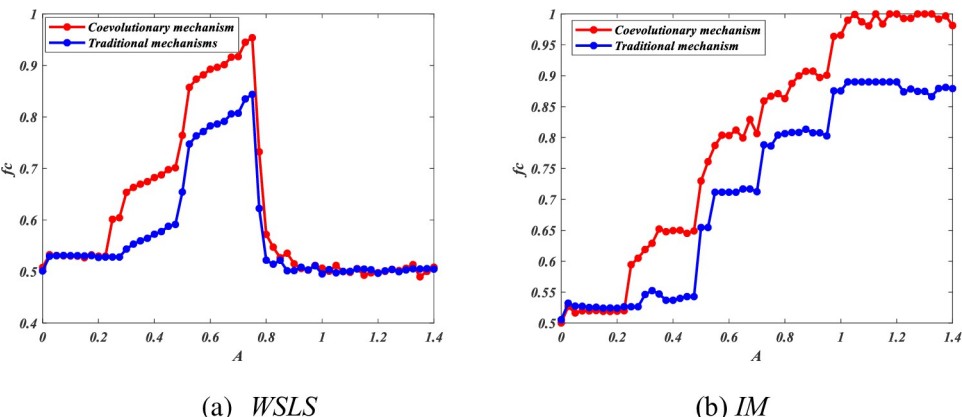

(a) *WSLS*                                                                (b) *IM*

**Fig 11. Cooperation level comparison between the co-evolutionary mechanism and traditional case, in which without mechanism, whether IM or WSLS, the co-evolutionary mechanism performs better than the traditional situation.** All results are obtained under *L* = 100. (a) *WSLS* and (b) *IM*.

cooperation levels. In the IM rule, high enhancement factor is not conducive to forming cooperation. Compared with the IM rule, the WSLS rule can better resist the invasion of defectors when aspiration level is low. By increasing aspiration, the IM rule has more advantages in enhancing cooperation. In addition, the heterogeneous will be beneficial to the evolution of cooperation. Lastly, we compared this mechanism with the traditional situation, and we found that whether IM or WSLS is involved, the co-evolutionary mechanism performs better than the traditional situation. Our research compensates for the shortcomings of existing studies and explains the effects of aspiration on cooperation emergence and maintenance based on the co-evolution of personal rationality. This mechanism provides a considerably detailed explanation to help us understand the phenomenon of cooperation depending on aspiration-driven in a real social system.

## Author Contributions

**Investigation:** Shounan Lu.

**Supervision:** Ge Zhu.

**Validation:** Jianhua Dai.

**Writing – original draft:** Shounan Lu.

**Writing – review & editing:** Ge Zhu.

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
