## [Decision Letter · Decision Letter 0]

9 Sep 2022

PONE-D-22-20446The willingness intensity and co-evolution of decision rationality depending on aspiration enhance cooperation in the spatial public goods gamePLOS ONE

Dear Dr. shounan,

Thank you for submitting your manuscript to PLOS ONE. After careful consideration, we feel that it has merit but does not fully meet PLOS ONE’s publication criteria as it currently stands. Therefore, we invite you to submit a revised version of the manuscript that addresses the points raised during the review process.

We look forward to receiving your revised manuscript.

Kind regards,

Xiaojie Chen

Academic Editor

PLOS ONE

Journal Requirements:

"This work was supported by national key research and development plan of China under grants nos. 2019YFB1406500."

"Unfunded studies

The author(s) received no specific funding for this work."

"NO authors have competing interests

Enter: The authors have declared that no competing interests exist."

6. PLOS requires an ORCID iD for the corresponding author in Editorial Manager on papers submitted after December 6th, 2016. Please ensure that you have an ORCID iD and that it is validated in Editorial Manager. To do this, go to ‘Update my Information’ (in the upper left-hand corner of the main menu), and click on the Fetch/Validate link next to the ORCID field. This will take you to the ORCID site and allow you to create a new iD or authenticate a pre-existing iD in Editorial Manager. Please see the following video for instructions on linking an ORCID iD to your Editorial Manager account: https://www.youtube.com/watch?v=_xcclfuvtxQ.

Reviewers' comments:

Reviewer's Responses to Questions

**Comments to the Author**

1. Is the manuscript technically sound, and do the data support the conclusions?

Reviewer #1: Partly

Reviewer #2: Yes

Reviewer #3: Yes

2. Has the statistical analysis been performed appropriately and rigorously? 

Reviewer #1: N/A

Reviewer #2: Yes

Reviewer #3: Yes

3. Have the authors made all data underlying the findings in their manuscript fully available?

Reviewer #1: No

Reviewer #2: Yes

Reviewer #3: Yes

4. Is the manuscript presented in an intelligible fashion and written in standard English?

Reviewer #1: No

Reviewer #2: No

Reviewer #3: No

5. Review Comments to the Author

Reviewer #1: In this work, Lu et al. introduced a game mechanism into the spatial public goods game where individuals’ rational emotions change synchronously with their aspiration. Concretely, the satisfaction level of individuals with the current payoffs will affect their rational decision-making level in the next round. They assumed that the intensity of personal subjective willingness to change the status quo is affected by the gap between aspiration and payoff. The greater the gap, the stronger the desire to change. In addition, the role of the strategy-updating rule in the evolution of cooperation has been studied. They found that the heterogeneous strategic update rule would benefit the evolution of cooperation.

While I find the initial idea behind the manuscript very interesting, I do not think that the study is of sufficient quality for publication.

First of all, the setting of the expected payoff of game players is unclear. In equation (2), the authors set that the expected payoffs of game players meet P_iA=A∙ K_i. Why this form? The authors should explain why this linear form is reasonable. Besides, in equation (4), W_i should be changed to w_i.

The second main problem is that the readability of the manuscript is very poor. Almost every page of the manuscript has grammatical problems, which makes reading very difficult.

The level of rationality is an important indicator for the authors to design this study. But I don't seem to see how the average rational level of the whole population will evolve.

Small comments:

In Figure 1 (c), what does data5 mean?

The caption of Figure 2, “The evolution process for A= A=0.3, 0.5, 0.7, 0.7, 0.9, respectively,” “(b) The evolution process for A=0.3, 0.5, 0.7, 0.7, 0.9, respectively, and r=7”. These descriptions do not correspond to those expressed in the Figure 2.

To conclude, I think that the idea behind the manuscript is very interesting and that the authors have made important steps to investigate this idea. However, I advise to rewrite the manuscript more carefully to be ready for publication.

Reviewer #2: This paper has proposed a spatial public goods game model where the individual rational sentiment and the intensity of willingness are changing depended on the difference of aspiration and payoff. The authors consider two types of strategy updating rules, stochastic “win-stay-lose-shift” rule (WSLS) and random imitation rule (IM). Through numerical simulations, they show that high value of enhancement factor is not conducive to cooperation in IM rules. Moreover, for small aspiration level, WSLS is more conducive to promoting cooperation than IM, while the situation is reverse for high payoff aspiration.

The topic of this paper is worth investigation. However, there are still some problems in this paper. These concerns that should be addressed before I can give my final recommendation. My detailed comments are as follows.

1.Eq.(1) can only describe the payoff that individual i obtains from the group centered on himself/herself, the authors need to check whether the results provided in this paper are based on the correct calculation method of accumulating payoff.

2.In the model section, it is mentioned that “when individuals are very dissatisfied with his/her returns, he/she will have a strong desire to change the status quo, and vice versa.”, while it is contrary to Eq.(4). Besides, ‘W_i(t)’ should be revised to ‘omega_i(t)’.

3.Visualization of all figures should be further improved.

4.Many sentences are ill-constructed. I hope that the authors can continue to check grammars, spelling and other problems in the next version of the manuscript.

5.In the introduction section, I suggest authors to mention four related works, i.e., “The role of noise in the spatial public goods game, JSTAT 2016”, “Public cooperation in two-layer networks with asymmetric interaction and learning environments, AMC 2019”, “Heterogeneous investments induced by historical payoffs promote cooperation in spatial public goods games, CSF 2020” and “Evolutionary public goods games on hypergraphs with heterogeneous multiplication factors, Acta Physica Sinica 2022”.

Reviewer #3: Comments on PONE-D-22-20446

Dear authors:

This paper proposes a spatial public goods game mechanism in which individuals' rational sentiment evolves synchronously depending on the difference between aspiration and payoff, and the intensity of their subjective willingness to change the strategies, which are dependent on the gap between aspiration and payoff. In general, this paper is quite interesting and helpful for us to understand the emergence of cooperation in social dilemmas. However, this paper still has some major problems needed to solve before being published in this journal.

1. It is the biggest problem for the authors is to improve their English expression and correct grammar mistakes. Actually, I found a lot of grammar mistakes in the main text, which make your study hard to understand. For example, the sentence, ‘rational sentiment is evolves’ in the Abstract is wrong, and the correct form should be ‘rational sentiment evolves’. I can understand that it is not easy for non-native speakers to write English papers. But to provide readers with a high-quality research, I suggest that the authors should ask native speakers for help to proofread your paper. If the resubmitted paper is still bad-writing, I will reconsider whether this paper is okay to be published on PLOS One.

2. I am confused about the realistic meaning of that the authors combined aspiration with rationality. Could you explain the meaning of the new mechanism you proposed and where it can be applied to solve social dilemma problems in the real world?

3. The mathematical symbols are misleading. For example, W in eq.4 means the willingness, but in eq.6, it represents the probability of updating strategy. The authors should carefully check the main text to avoid similar small mistakes.

4. I doubt the correctness of the simulation results. Because the simulation results in Fig.1 show that the cooperation level is higher as r is lower. This conclusion does not make sense and is contrary to that in obvious studies on the public goods game.

5. In the main text, the authors investigate the cooperation dynamics under WSLS and IM, respectively. But this is not enough to verify the conclusion, namely, ‘The willingness intensity and co-evolution of decision rationality depending on aspiration enhance cooperation in the spatial public goods game’. To provide more persuasive evidence to verify the main conclusion, I suggest the authors introduce a benchmark, which is not influenced by any mechanism. Then, by comparing the cooperation dynamics of the new mechanism proposed with the benchmark, it is easy to verify whether cooperation is promoted by the proposed mechanism.

6. There are some wrong formats of literature in the Reference part, such as [14], [27]. The authors should carefully check and avoid unnecessary errors.

6. PLOS authors have the option to publish the peer review history of their article (what does this mean?). If published, this will include your full peer review and any attached files.

Reviewer #1: No

Reviewer #2: No

Reviewer #3: No

---

## [Decision Letter · Decision Letter 1]

3 Nov 2022

PONE-D-22-20446R1The willingness intensity and co-evolution of decision rationality depending on aspiration enhance cooperation in the spatial public goods gamePLOS ONE

Dear Dr. Dai,

Thank you for submitting your manuscript to PLOS ONE. After careful consideration, we feel that it has merit but does not fully meet PLOS ONE’s publication criteria as it currently stands. Therefore, we invite you to submit a revised version of the manuscript that addresses the points raised during the review process.

We look forward to receiving your revised manuscript.

Kind regards,

Xiaojie Chen

Academic Editor

PLOS ONE

Journal Requirements:

Reviewers' comments:

Reviewer's Responses to Questions

**Comments to the Author**

1. If the authors have adequately addressed your comments raised in a previous round of review and you feel that this manuscript is now acceptable for publication, you may indicate that here to bypass the “Comments to the Author” section, enter your conflict of interest statement in the “Confidential to Editor” section, and submit your "Accept" recommendation.

Reviewer #1: (No Response)

Reviewer #2: All comments have been addressed

Reviewer #3: All comments have been addressed

2. Is the manuscript technically sound, and do the data support the conclusions?

Reviewer #1: Yes

Reviewer #2: Yes

Reviewer #3: Yes

3. Has the statistical analysis been performed appropriately and rigorously? 

Reviewer #1: Yes

Reviewer #2: Yes

Reviewer #3: Yes

4. Have the authors made all data underlying the findings in their manuscript fully available?

Reviewer #1: Yes

Reviewer #2: Yes

Reviewer #3: Yes

5. Is the manuscript presented in an intelligible fashion and written in standard English?

Reviewer #1: Yes

Reviewer #2: Yes

Reviewer #3: No

6. Review Comments to the Author

Reviewer #1: The authors have substantially revised the manuscript and incorporated my suggestions.

I only have two minor comments.

It is known for the evolutionary game that incentive strategies can significantly affect

the evolution of cooperation (see for example, Chen et al., Journal of the Royal Society Interface 12.102 (2015): 20140935; Physical Review E 92.1 (2015): 012819; Liu and Chen, Applied Mathematics and Computation 425 (2022): 127069;Mathematical Models and Methods in Applied Sciences 29.11 (2019): 2127-2149;Journal of the Royal Society Interface 19.188 (2022): 20210755). It would be very meaningful if the authors add these works.

Characters in the manuscript should be italicized, such as R_i in eq. 5 and eq. 6.

Reviewer #2: The revised manuscript seems enough to persuade me. I would like to recommend it for publication in PLOS ONE.

Reviewer #3: 1. The English expression has been improved, but there are still many grammar mistakes. It is suggested that the author seek professional researchers to modify the grammar.

2. Does A evolve or stay the same during the evolutionary process? And is A heterogeneous among different people? Please elaborate on this question in the revised manuscript.

3. Under the different strategy update rules of WSLS and IM, what new discoveries can be provided by the distribution of strategies, willingness, and rationality on the square lattice? This seems not to be elaborated in the main text.

7. PLOS authors have the option to publish the peer review history of their article (what does this mean?). If published, this will include your full peer review and any attached files.

Reviewer #1: No

Reviewer #2: No

Reviewer #3: No

---

## [Author Response · Author response to Decision Letter 1]

26 Nov 2022

Dear reviewer:

Thank you for your decision and constructive comments on my manuscript. We have carefully considered the suggestion of Reviewer and make some changes. We have tried our best to improve and made some changes in the manuscript.

The yellow part that has been revised according to your comments. Revision notes, point-to-point, are given as follows:

To Reviewer #1: 

Review comments 1: It is known for the evolutionary game that incentive strategies can significantly affect the evolution of cooperation (see for example, Chen et al., Journal of the Royal Society Interface 12.102 (2015): 20140935; Physical Review E 92.1 (2015): 012819; Liu and Chen, Applied Mathematics and Computation 425 (2022): 127069; Mathematical Models and Methods in Applied Sciences 29.11 (2019): 2127-2149; Journal of the Royal Society Interface 19.188 (2022): 20210755). It would be very meaningful if the authors add these works.

Reply to comments: 

Thank reviewer 1 for his comments on the revision. According to the comments, we cited relevant literature to improve the practical significance of the article.

Review comments 2:

Characters in the manuscript should be italicized, such as R_i in eq. 5 and eq. 6.

Reply to comments: 

Thank the reviewers for their careful review. We have adjusted and marked.

Such as eq.3 : ; eq.5: and eq.6: 

To Reviewer #3: 

Review comments 1: The English expression has been improved, but there are still many grammar mistakes. It is suggested that the author seek professional researchers to modify the grammar.

Reply to comments 1: 

According to the suggestion, we have sought help from professors in the research neighborhood and made modifications.

Review comments 2: Does A evolve or stay the same during the evolutionary process? And is A heterogeneous among different people? Please elaborate on this question in the revised manuscript.

Reply to comments 2: 

The Aspiration level A will stay the same during the evolutionary process, and we have explained this problem in detail and identified it in the revised version.

Review comments 3: Under the different strategy update rules of WSLS and IM, what new discoveries can be provided by the distribution of strategies, willingness, and rationality on the square lattice? This seems not to be elaborated in the main text.

Reply to comments 3: 

For the distribution of strategies, willingness, and rationality on the square lattice and the effect of willingness, and rationality on cooperation, we have elaborated in the revised version and marked. The details are as follows.

Fig. 3 shows the distribution of the population’s strategies in the time evolution process for several time steps from a micro perspective. When the aspiration is small, A=0.3, both cooperators and defectors are satisfied with returns, as shown in Fig.4, they have a lower willingness to change the status quo. However, there are still a few individuals close to the cooperators cluster for a free ride on the whole population. Moreover, cooperators form clusters to resist the invasion of defectors by obtaining stable benefits. Clusters gradually expand with an increase in aspiration. Moreover, by analyzing the distribution of cooperation strategy and the corresponding rational emotional state of individuals in the stable state for different aspirations A, completely rational individuals always maintain a high willingness to update their strategies, that is, they always hope to maximize their own benefits by constantly adjusting their strategies in which cooperation is difficult to maintain. The reasonable explanation that some rationality is conducive to cooperation is that the decision-making process of individuals not only considers the maximization of benefits, but also considers the external environment and decision-making conditions, which leads to the maintenance of cooperation.

---

## [Decision Letter · Decision Letter 2]

21 Dec 2022

The willingness intensity and co-evolution of decision rationality depending on aspiration enhance cooperation in the spatial public goods game

PONE-D-22-20446R2

Dear Dr. Dai,

We’re pleased to inform you that your manuscript has been judged scientifically suitable for publication and will be formally accepted for publication once it meets all outstanding technical requirements.

Kind regards,

Xiaojie Chen

Academic Editor

PLOS ONE

Review Comments to the Author

Reviewer #1: In their revised version, the authors addressed all points which I raised in a satisfactory manner. As far as I am concerned the paper is worth publishing and is essentially ready to go.

Reviewer #3: (No Response)

---

## [Editor Report · Acceptance letter]

3 Jan 2023

PONE-D-22-20446R2 

Willingness intensity and co-evolution of decision rationality depending on aspiration enhance cooperation in the spatial public goods game 

Dear Dr. Dai:

I'm pleased to inform you that your manuscript has been deemed suitable for publication in PLOS ONE. Congratulations! Your manuscript is now with our production department. 

Kind regards, 

on behalf of

Professor Xiaojie Chen 

Academic Editor

PLOS ONE